# Optically Induced Field-Emission Source Based on Aligned Vertical Carbon Nanotube Arrays

**DOI:** 10.3390/nano11071810

**Published:** 2021-07-12

**Authors:** Mengjie Li, Qilong Wang, Ji Xu, Jian Zhang, Zhiyang Qi, Xiaobing Zhang

**Affiliations:** School of Electronic Science and Engineering, Southeast University, Nanjing 210096, China; 230189115@seu.edu.cn (M.L.); northrockwql@seu.edu.cn (Q.W.); xu@seu.edu.cn (J.X.); contact_zhangjian@163.com (J.Z.); qizhiyang2009@163.com (Z.Q.)

**Keywords:** photo-electron emission, optically induced field emission, field emission, VCNTAs, photosensitivity

## Abstract

Due to the high field enhancement factor and photon-absorption efficiency, carbon nanotubes (CNTs) have been widely used in optically induced field-emission as a cathode. Here, we report vertical carbon nanotube arrays (VCNTAs) that performed as high-density electron sources. A combination of high applied electric field and laser illumination made it possible to modulate the emission with laser pulses. When the bias electric field and laser power density increased, the emission process is sensitive to a power law of the laser intensity, which supports the emission mechanism of optically induced field emission followed by over-the-barrier emission. Furthermore, we determine a polarization dependence that exhibits a cosine behavior, which verifies the high possibility of optically induced field emission.

## 1. Introduction

Optically induced field-emission has drawn extensive research attention in next-generation ultrafast electron diffractive imaging and spectroscopy [1], compact coherent x-ray sources [2], and attosecond research [3,4]. In the investigation of a hybrid emission regime for ultrafast optically induced field-emission, a compelling trend is the employment of increasingly exotic nanomaterials. This is because the field enhancements stem from the local geometry and the sub-wavelength confinement of the optical field [5,6]. The potentially lower work function for such nanomaterials is also a favorable factor that reduces the need for the strong optical field that is normally required [7], and the strong-field photoelectron emission from single metal nanostructures has resolved the transition between multiphoton photoemission and optical field emission [8]. Other work has focused on few-cycle femtosecond (fs) laser pulses acting on a sharp tungsten tip, followed by measuring the energy of the emitted electrons [9]. In general, many works on multiphoton absorbing nanostructures and strong-field emission cathodes focus on single metal tips [8,10,11,12], a single nanomaterial [5,13,14,15,16], and an array of nano-sharp high-aspect-ratio silicon columns [17]. However, the high costs of traditional materials and the difficult fabrication process [18,19,20,21], which causes the tip to ‘burn out’ in a strong field, are barriers to the practicability of their material. Therefore, it is urgent to discover novel kinds of nanomaterial with favorable properties, such as the ability to be prepared in large quantities (using a simple preparation process) and robust electron emission abilities under laser irradiation. Possible nanomaterials that meet these requirements are carbon nanotubes (CNTs) [22,23,24,25,26]. However, when using CNTs with a one-dimensional morphology as an electron source, the inherent strong exciton resonance and high thermal conductivity may result in a high damage threshold [27,28,29,30,31,32,33,34]. Compared with low work-function materials [35,36,37], CNTs show excellent photon absorption efficiency [38,39]. However, a thorough exploration of the femtosecond laser-induced field-emission mechanism based on CNTs arrays is lacking. Besides, combining the advantages of CNTs arrays for field emission with optically induced field-emission, we can investigate the optimum conditions of emission that ensure highly monochromatic ultrafast laser-triggered electron emission, thereby improving its practical value as an electron source.

In this paper, vertical carbon nanotubes arrays (VCNTAs) were used as the emitting cathode. The cathodes exhibited low turn-on field (1.2 V/μm) and 2% fluctuation of emission stability. It is also noteworthy that the emission current was strongly dependent on the polarization angle of the incident laser, which exhibited an obvious cosine behavior, verifying that optical field emission rather than thermal-induced emission was occurring. The experimental results clearly showed the three electron emission processes for the VCNTAs with laser irradiation. Therefore, the VCNTAs generate high-throughput electron beams and they show significant light sensitivity to the ultrafast laser.

## 2. Experimental

The VCNTAs were fabricated by the chemical vapor deposition (CVD) technique [40] on a commercial highly doped silicon substrate (n-type, the RDMICRO in Suzhou) with a conductivity of 0.003 Ω·cm. First, the substrate was patterned using SPR220-7.0 UV resist and an MA6 ultraviolet lithography device. Then, the patterned substrates were used to sequentially deposit aluminum (20 nm) and Fe (10 nm) catalyst layers via magnetron sputtering. Finally, the substrates were placed into a quartz tube reactor (length of 120 mm, inner diameter of 100 mm) and mixed argon (Ar, 99%, 100 sccm) in a tube furnace. Thje furnace was heated to 700 °C and methane (CH_4_, 99% 200 sccm) and hydrogen (H_2_, 95%, 200 sccm) were passed through the tube reactor for 10 min at a pressure of 4000 Pa, followed by cooling the CVD system in Ar to room temperature. The side and top views of the fabricated VCNTAs were observed via a (Quanta 200 FEI) scanning electron microscope (SEM) images of which are shown in Figure 1a,b. The multiple-walled CNTs in the array are uniform, dense, well-aligned, and perpendicular to the substrate. The CNTs have diameters of ~30 nm and dimensions characteristic of a VCNTA (L_1_ = L_2_ = 20 μm, H = 43 μm). An image of these, as seen by a (JEOL-2010F) transmission electron microscope (TEM), is presented in Figure 1c. This image shows the characteristics of the lattice fringe. A typical He & I ultraviolet photoelectron spectrum (UPA, an ESCALAB 250Xi electron spectrometer (Thermo Fisher Scientific, Waltham, MA, USA)) in the binding energy scale is shown in Figure 1d for the CNTs. The composition analysis was performed by energy-dispersive X-ray analysis (EDS). A sample of the VCNTAs was confirmed by EDS analysis (Figure 1e), which exhibited the expected signal peaks of C and Si.

The experimental setup needed a high vacuum environment and a relevant light path. The test sample consisted of a cathode base, a sheet metal anode, and a ceramic gasket that separated the cathode from the anode. The cathode was positioned about 0.2 mm away from the anode, which had a 1 mm diameter hole (as shown on the right-hand side of Figure 2). Then the test setup was connected to a high vacuum of 4 × 10^−6^ Pa. A Keithley 6487 source measurement unit was used to apply bias voltages and collect the emission current. To ensure measurement of the optical-driven electron field emission, we built the light path, as shown on the left side of Figure 2. The VCNTAs are front illuminated by a 150 fs/1000 Hz fs-laser of a central wavelength of 650 nm with the laser excitation power varied using a variable attenuator from approximately 0 mW to 90 mW. A laser power meter (Spectra-Physics 407A) was used to measure and calibrate the laser power, and the power density range of the mono-pulse was determined. The laser was directly injected into the cathode through a quartz window. The power density ranged from 0 to 0.306 TW/cm^2^, which was calculated from the following formula for laser power:(1)P0=Ulaserft×1S
where $U_{laser}$ is the laser energy, *f* is the laser frequency, *S* is the surface area of the laser beam and *t* is irradiation time. The laser was focused on a light spot with a diameter of 1 mm on the VCNTAs.

## 3. Results and Discussion

In order to investigate the optically induced field-emission characteristics, J-E curves were analyzed by using the different laser powers as shown in Figure 3a,b. We observed a rapid rise of the photocurrent at a lower bias (<20 V) due to the space-charge effect in which the photons and the ionized electrons form an electron cloud distribution around the cathode, which is attracted to the anode as the bias increases [41]. The electron emission properties of the VCNTAs were highly dependent on both the laser power and the bias voltage. At an electric field of 1.43 V/μm, the VCNTAs obtained an emission current of ~1 μA/cm^2^ without laser irradiation. This electric field is regarded as turn on field. When the input laser power was increased to 45 mW, the turn on electric field of the VCNTAs decreased to 1.2 V/μm. The turn-on voltage of the VCNTAs was 1.28 V/μm at the laser powers up to 90 mW. This is because the strong light field effects dominate the vacuum barrier, and the multiphoton light emission channel becomes saturated at a higher intensity [6]. Moreover, we measured the emitter-current as a function of the bias-voltage when the laser power was switched from 0 to 90 mW. In both cases, the data fits into the Fowler–Nordheim equation [42], in which *J* is related to the tunnel current density, *F* is the local hybrid field strength, and Փ is the effective work function:(2)J=e3F28πhϕt2(ω)E[8π2mϕ323heFυ(ω)]
(3)$$I=2\pi R^{2}J$$
where *I* is the emitter current, *R* is the radius of the cathode emitter area, and *F* is the composite field, which is composed of the electric field and optical wave field. Then, the field enhancement factor (*β*) of CNTs can be calculated from the slope (*S*) of the linearized F-N data, using the transformed to Equations (2) and (3).
(4)$$\ln\left(\frac{I}{V^{2}}\right)=-\frac{b\phi^{\frac{3}{2}}d}{\beta }\left(\frac{1}{V}\right)+\ln\left(\frac{Aa\beta^{2}}{\phi d^{2}}\right)$$
(5)$$s=\frac{-b\phi^{\frac{3}{2}}d}{\beta }$$
where *Փ* is the work function, *d* is the distance between cathode and anode, *a* = 1.56 × 10^−6^, *b* = 6.83 × 10^7^. Taking the lowest turn-on field as the critical boundary surface, it divides the electric field into two parts as shown in Figure 3b. Combined with the calculation of Formula (5), the values of s were 19.23 (no-laser), 18.96 (14 mW), 20.54 (45 mW), and 21.69 (90 mW). It was shown that the different laser powers have a weak influence on the field enhancement factor and proves that the electron emission mechanism is in total accordance with the formula F-N [43] under the high electric field. Moreover, the conclusion also indicates that the main mechanism of the electron emission is field emission under the high electric field.

Further supporting evidence for the optically induced field-emission is provided by the input light polarization dependence of the photocurrent as shown in Figure 3c. Using a polarizing plate, the incident laser polarization angle was varied between the angles of 0° and 360°, which clearly showed that the emission current has a polarization dependence. This curve also largely excludes the possibility of a thermally induced field emission [10]. Thus, we can infer that electron emission is rapid on the input of the laser pulse, and the possibility of any thermal emission mechanisms associated with laser-induced heating of the tip can be ruled out [44]. For a bias voltage of 300 V, there is more than ~90% chance that the emitted electrons derive from the optical input field. When excited by the ultrafast laser pulses, electrons may be excited to nonequilibrium states by obtaining energy from both photons and thermal (laser-heating) effects. In the case of photon-driven excitation, the time scale of the general electron pulse is the same as that of the laser pulse [45]. In the case of thermally driven excitation, time frames of >100 fs are required to transfer sufficient thermal energy to the local electron population.

To obtain the relationship between the electron emission and the input laser power, which is mainly laser-induced when below the turn-on voltage, the following uses the Fowler−DuBridge model [46] in which the traces are fit with a polynomial:(6)$$I=\sum C_{n}P^{n}$$
where *n* is the order of the photon process, P is the average laser power, and *C_n_* is a fitting parameter. Figure 3d shows, at a bias voltage of 300 V, the electron photoemission currents as a function of the laser intensities. At 0–45 mW we obtained *n* = 0.599 and at 45–90 mW *n* = 0.2979, which are illustrated by the red and blue dashed lines (intentionally offset for clarity in the log-log scale) in Figure 3d. We observed continuous transitions between power laws of different orders. Beyond 45 mW, the slope of the current becomes smaller. This is because the multiphoton emission channel is saturated at a higher intensity. In such cases, the strong light field effects dominate the vacuum barrier [41].

A hybrid emission regime for strong-field above-threshold photoemission has been widely studied [7], but the process for femtosecond laser-assisted field-emission with varying input electric fields has lacked a thorough exploration [26]. It has been shown that VCNTAs can operate via three processes as the electric field increased [9,47,48], (a) photo-electron emission, (b) optical induced field emission, and (c) field emission. Under these regimes, electrons are first excited from their original energy level to a higher energy-level state by absorbing thermal or photon energy. Then, they face a much narrower tunneling barrier, meaning that the tunneling probability is greatly increased, which results in a highly enhanced emission current.

The relationship between emission current and the illumination time in the bias voltage range from 100 V to 400 V and the laser power range from 0 to 90 mW is as follows: The current fluctuation depends on the laser power when the bias voltage is less than 300 V as exhibited in Figure 5a,b. In such conditions, the active process is photo-electron emission, as shown in Figure 4a, which involves excitation of the surface of the VCNTAs by a large number of photons, so that electrons overcome the vacuum barrier inside the material. When the bias voltage is increased, the vacuum barrier is compressed and the electrons inside the material can escape more easily from the energy barrier. The fluctuating current here is quite remarkable since a lot of active electrons can tunnel out of the material with a strong photo-electric field [44]. The process is the optically induced field-emission shown in Figure 4b. In this situation, the electrons are excited by the laser into an intermediate state, and the electrons can then tunnel into the vacuum via the potential barrier at the tip of the material that is compressed by the applied DC voltage. As shown in Figure 5c, the current fluctuations still exist, which is related to the optical modulation with the bias voltage at 1.5 V/μm. Simultaneously, the maximum value of the emission current is 6.57 × 10^−7^ A for a laser power of 45 mW, which is more than 13 times the current without the irradiation condition at the same bias voltage. Finally, when the bias voltage is high enough, the energy barrier is compressed which allows the electrons to tunnel to the vacuum. By comparing the results of Figure 5c, the fluctuations in the current reflect the dominant effects of the electric field and optical field on the electric emission of the carbon nanotubes [49]; in these cases, the optical field instantaneously fluctuates the energy barrier [50]. When the bias voltage is up to 400 V, the photo-current is almost nonexistent, as shown in Figure 5d. Here, the bias voltage is too high, the energy barrier is completely compressed, so that the electrons can freely tunnel to the vacuum. It is also shown that electron emission from the carbon nanotubes has three progressive pathways via optical and electronic fields.

The stability test is shown in Figure 6a. For a bias voltage of 300 V and a laser power of 45 mW, the emission current was 2.3 × 10^−7^ A and the emission variability of the cathode of VCNTs was less than 2% during the 2400 s. With the irradiation time prolonged, the emission current appeared to fluctuate increasingly, the current fluctuations were extended further (the average current increased). The light excitation was stopped at 1200 s, as shown in Figure 6c. This is because the fs-laser peak power density is too high and the cathode could be ruined in these circumstances. Combined with the SEM image of the cathode surface showing its deformation, as shown in Figure 6d,e, the top of the carbon nanotubes array may show traces of break up.

## 4. Conclusions

To summarize, vertical carbon nanotubes, acting as emission cathode, can generate high-throughput electron beams with high photosensitivity under the irradiation of an ultrafast laser. The vertical carbon nanotubes arrays (VCNTAs) were used as the cathode for optical field emission, in combination with a high applied electric field and laser illumination, allowing us to modulate the electron emission with a fs-pulse laser. The cathodes exhibited a current of 6.57 × 10^−7^ A at 1.2 V/μm, with a 2% fluctuation of the electron emission. The VCNTAs generate high-throughput electron beams and they exhibit significant light sensitivity to the ultrafast laser. It is possible that they may find applications in high-brightness electron beam technology and ultra-fast imaging.

## Figures and Tables

**Figure 1 nanomaterials-11-01810-f001:**
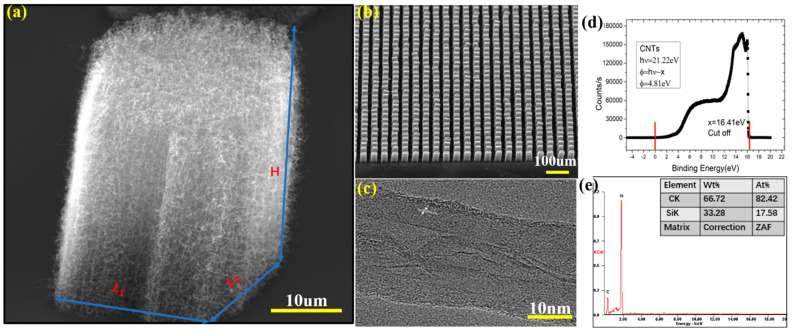
SEM images of (**a**) a cluster of nanotubes, and (**b**) a top view of the aligned CNT-array sample, (**c**) (HR)TEM images of the single CNT, (**d**) A work function of 4.81 eV is determined from the secondary emission onset in ultraviolet photoelectron spectroscopy, (**e**) EDS test of the aligned CNT-arrays sample.

**Figure 2 nanomaterials-11-01810-f002:**
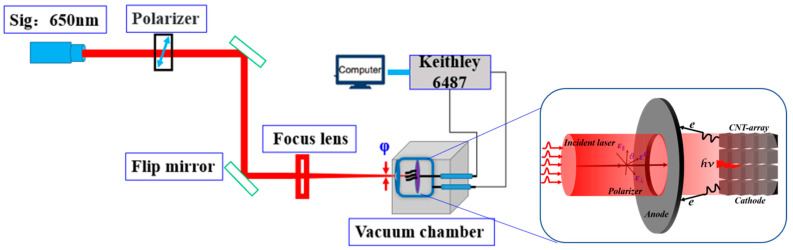
A schematic of photoemission configuration.

**Figure 3 nanomaterials-11-01810-f003:**
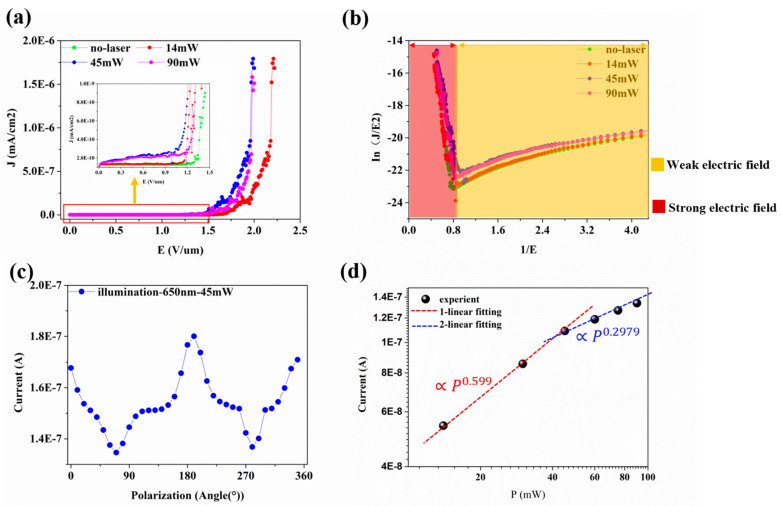
(**a**) J-E curves for an excitation laser wavelength of 650 nm at selected average values of the input laser power. The inset is enlarged for the y-axis values between 0 to ~1 × 10^−9^ mA/cm^2^; (**b**) The Fowler-Nordheim (F–N) plots. (**c**), Photocurrent curves dependent on the incident light polarization, given at a 300 V bias voltage and the 14 mW laser power for 650 nm excitation. (**d**) I-P curves at selected bias voltages for 650 nm excitation. The dotted lines represent the fitting curves.

**Figure 4 nanomaterials-11-01810-f004:**
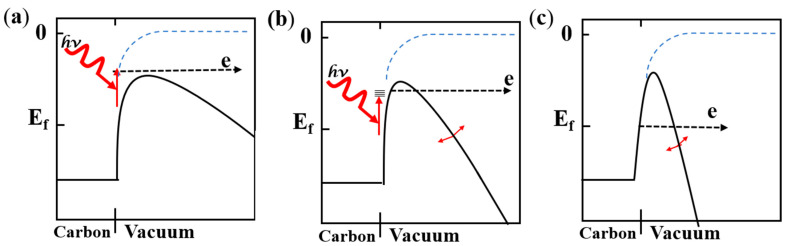
Energy diagrams for (**a**) photo-electron emission, (**b**) optically induced field emission, and (**c**) field emission.

**Figure 5 nanomaterials-11-01810-f005:**
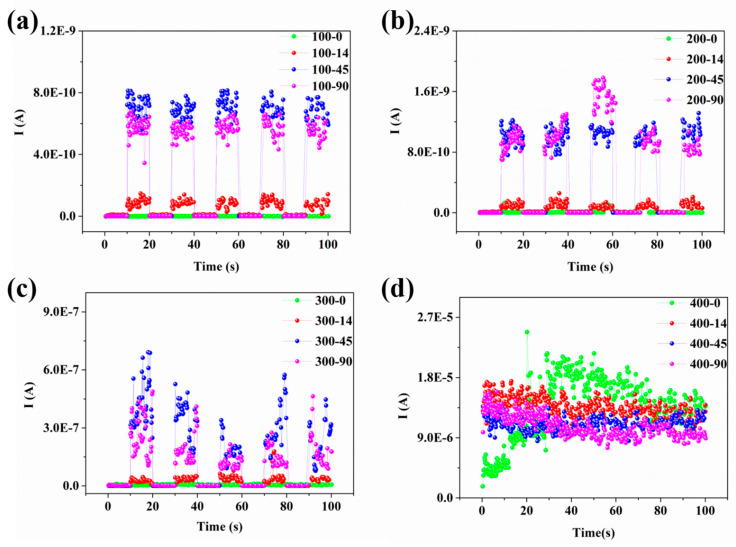
The variation of emission current of the VCNTAs under the different laser power and bias voltage, when the laser is switched on and off. Different colors in each figure represent different excitation laser power, (**a**) at a bias of 100 V, (**b**) at a bias of 200 V, (**c**) at a bias of 300 V, and (**d**) at a bias of 400 V.

**Figure 6 nanomaterials-11-01810-f006:**
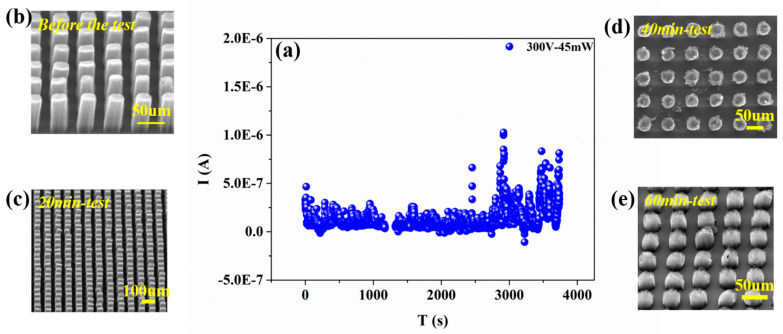
The stability of electron emission, (**a**) the fluctuation of emission current with the period time. The morphology of VCNTs (**b**) before the test, (**c**) after being irradiated for 20 min, (**d**) after being irradiated for 40 min, and (**e**) after being irradiated for 60 min.

## Data Availability

Data are contained within the article.

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
