# Peer review of "Optically Induced Field-Emission Source Based on Aligned Vertical Carbon Nanotube Arrays"

_nanomaterials, 2021, doi:10.3390/nano11071810_

Round 1

Reviewer 1 Report

The manuscript entitled “Optically Induced Field-emission Source based on aligned Vertical Carbon Nanotubes Arrays” reports an experimental study in which the performances of vertical carbon nanotube arrays as emitting cathodes are investigated. Based on the results obtained, the authors attest that VCNTAs can generate high-performance electron beams with high photosensitivity under the irradiation of an ultra-fast laser. It is possible to operate through three different electron emission processes as the electric field increases.

The research topic is interesting and relevant to the purpose of the journal. The experimental work was carried out in depth and the results are discussed comprehensively. However,in reading the manuscript I found some parts that I suggest to the authors to review. In light of the above, I believe that the work can be considered for its publication but after a minor revision.

Here are some suggestions for the authors:

*) The experimental part should be improved.

*)Line 64. Although the bibliographic note [31] is reported, I suggest giving more information on the preparation of VCNTAs. This would make the manuscript more complete.

*)The  caption of figure 6 is incomplete.

*) Conclusions could be improved. The last sentence should also be clarified: "It is possible that on the application of high-dose electron beam technology and rapid imaging".

Reviewer 2 Report

The authors describe optically induced field-emission on vertically aligned carbon nanotube arrays. The research subject is of broad interest and the report will be welcome by a large scientific community, however improvements will need to be done in the English.

The abstract must be improved.

Introduction

The authors must improve the introduction adding numbers for thermal conductivity of CNTs and silicon, explain the reason that this property is so important.

The authors must compare the work-function of different materials.

To what extend chemical stability of CNTs is important.

Experimental

It is not clear if the Si used was commercial, how was it synthesized?

It is important to give an extended explanation of the CNT growth? What gases were used? At what temperature the catalysts were heated before growth? What was the pressure and growth temperature?

Please describe the parameters used for UPS analysis of the sample and determination of the work-function? What type of spectrometer was used? Energy resolution? Bias applied? Number of samples analyzed?

What type and brand of the equipment used for EDS.

In lise 96 it is not clear what is the mentioned light spot.

Results and Discussion

Figure 3 must be better explained.

Line 168, it is not clear that vertically aligned carbon nanotubes can operate via three processes, the explanation must be extended. It is important to compare with other reports and materials.

It is not clear the stability of vertically aligned carbon nanotubes? How many samples were analyzed? Are the measurements repeated? Can error bars be added to figure 3 C and 3D.

Round 2

Reviewer 2 Report

The authors answered the questions. The manuscript should be revised carefully but it can be accept after text editing.
